# Cold-programmed shape-morphing structures based on grayscale digital light processing 4D printing

Liang Yue [1,4], Xiaohao Sun [1,4], Luxia Yu[1], Mingzhe Li [1],
S. Macrae Montgomery [1], Yuyang Song[2], Tsuyoshi Nomura[3], Masato Tanaka[2] &
H. Jerry Qi [1] ✉

Shape-morphing structures that can reconfigure their shape to adapt to diverse tasks are highly desirable for intelligent machines in many interdisciplinary fields. Shape memory polymers are one of the most widely used stimuli-responsive materials, especially in 3D/4D printing, for fabricating shape-morphing systems. They typically go through a hot-programming step to obtain the shape-morphing capability, which possesses limited freedom of reconfigurability. Cold-programming, which directly deforms the structure into a temporary shape without increasing the temperature, is simple and more versatile but has stringent requirements on material properties. Here, we introduce grayscale digital light processing (g-DLP) based 3D printing as a simple and effective platform for fabricating shape-morphing structures with cold-programming capabilities. With the multimaterial-like printing capability of g-DLP, we develop heterogeneous hinge modules that can be cold-programmed by simply stretching at room temperature. Different configurations can be encoded during 3D printing with the variable distribution and direction of the modular-designed hinges. The hinge module allows controllable independent morphing enabled by cold programming. By leveraging the multimaterial-like printing capability, multi-shape morphing structures are presented. The g-DLP printing with cold-programming morphing strategy demonstrates enormous potential in the design and fabrication of shape-morphing structures.

Shape-morphing structures that can change their configurations to adapt specific functionalities in response to external stimuli have led to advances for intelligent devices[1,2], from soft robotics[3–5], deployable systems[6,7], wearable devices[8], to shape-shifting antennas[9,10], sensors[11], and actuators[12,13]. A shape-morphing system often uses the composite design to achieve the capability of shifting between two or more prescribed configurations. One key factor in realizing these systems'

potential is the effective fabrication methods. Due to its great manufacturing flexibility, additive manufacturing (AM) or three-dimensional (3D) printing is a promising approach. Recently, merging 3D printing with stimuli-responsive materials led to the birth of 4D printing, with time as the 4th dimension[14]. Ever since the first report of 4D printing, it has been utilized to fabricate shape-morphing structures[15,16]. Among different stimuli-responsive materials used in 4D printing, shape

[1]The George W. Woodruff School of Mechanical Engineering, Georgia Institute of Technology, Atlanta, GA 30332, USA. [2]Toyota Research Institute of North America, Toyota Motor North America, Ann Arbor, Michigan 48105, USA. [3]Toyota Central R&D Laboratories, Inc., Bunkyo-ku, Tokyo 112-0004, Japan. [4]These authors contributed equally: Liang Yue, Xiaohao Sun. ✉e-mail: qih@me.gatech.edu

memory polymers (SMPs) are one of the most extensively explored because they can be programmed into many different shapes after printing and then recover the original shape under external stimuli, such as temperature [14,17–19].

In SMP applications, to achieve a shape memory cycle, an SMP needs to go through a programming and a recovery step. The programming step is typically conducted through a hot-programming process: it is heated to a temperature above a transition temperature (for example, glass transition temperature for amorphous polymers) and is then deformed. The temperature is then lowered while the deformed shape is held. To recover the permanent shape, the SMP is heated above the transition temperature[20–22]. Because SMPs are soft at the temperature above the transition temperature, hot-programming has the advantage of low programming force. However, it also has obvious disadvantages. It is typically conducted by heating the entire SMP structure (global heating) in a hot-water bath or oven, deforming it, and then cooling it while holding the deformed shape. This would increase the overall energy cost and make local programming very difficult. To achieve local programming, one would need to locally apply loads and maintain those loads during cooling or design local features so that the globally applied deformation could generate local deformation, such as the hinges in printed origami[23,24]. The latter has the drawback that the globally applied deformation would program all local features. Another approach to achieve local programming is to use localized heating, which requires additional hardware, such as embedded heaters, or using photothermal effects. It should be noted that this dilemma of global heating and local programming is common in most active structures using stimuli-responsive materials, such as liquid crystal elastomers and hydrogels[25,26]. Another approach, albeit less explored, is cold-programming, which deforms the SMP into the programmed shape at low temperatures without the heating-cooling cycle[27–31]. Cold-programming uses the principle that the relaxation time in polymers is a function of not only temperature but also stress[32–34]. At high stress, the relaxation time is significantly reduced, and the polymer exhibits plastic deformation, which is recoverable upon heating. Cold-programming has the advantage of room temperature programming, which can be conducted easily and locally. However, it deforms the material in a glassy state and thus requires a much larger force. It is also not always feasible as the material may break or be damaged. Cold-programming by deforming the SMP beyond yielding without fracture requires the SMP to be ductile enough to bear the large strain at low temperatures[35,36]. Because of such limitations, cold-programming has not been widely studied. However, the advantage of easy programming still makes it an attractive candidate for 4D printing.

In this work, we present a grayscale digital light processing (g-DLP) based single-vat multi-property printing platform for 4D printing cold-programmed shape-morphing structures[37,38]. With the rationally designed ink, the g-DLP platform can print materials ranging from ductile glassy thermoset to highly stretchable organogel with tunable elasticity and glass transition temperatures using a single vat of resin[39,40]. Furthermore, ductile glassy thermosets can be used as cold-programming SMPs. With this feature, we can fabricate cold-programmable SMP structures with heterogeneous material distributions. We design a bilayer-based hinge with glassy fibers embedded in a rubbery matrix as a cold-draw programmable unit for shape morphing. The optimized hinge design can be easily deformed with a small force and bear large strain for tunable deformation. Complex cold-programmable shape-morphing structures can be designed and printed with variable prescribed configurations encoded through the distribution and direction of the modular-designed hinges. Moreover, we are able to print the hinges with different thermomechanical responses to realize smart morphing structures that can change to multiple configurations. The cold-programmable

4D printing thus represents a new approach for fabricating shape-morphing structures for various applications.

## Results

### Single-vat g-DLP printing of materials with multi-property

A single-vat multi-property g-DLP printing approach is developed as the platform for 4D printing shape-morphing structures, as illustrated in Fig. 1A. The images encoded locally modulated UV light intensity (grayscale) for each pixel (50 μm) are projected onto the ink-vat window in a bottom-up DLP printer to solidify the UV curable resin. Light intensity for each pixel varys and yields different monomer conversions. With the rationally designed resin materials for g-DLP printing, the dim light (low light intensity or high grayscale) region prints a soft and rubbery organogel with a low degree of curing (DoC), whereas the bright region prints a glassy thermoset with a high DoC. The g-DLP ink consists of isobornyl acrylate (IBOA) and 2-hydroxyethyl acrylate (2-HEA) as the stiff and soft chain builders, respectively, and an aliphatic urethane diacrylate (AUD) as the crosslinker[39] (Fig. 1B). This ink formula features multiple hydrogen bonding donor and acceptor moieties, leading to abundant intermolecular hydrogen bonds. As a result, at the low DoC, the printed material forms an organogel with exceptional softness (Young's modulus of approximately 0.016 MPa) and stretchability (up to approximately 1500%). At the high DoC, the covalent network with stiff IBOA blocks raises the glass transition temperature ($T_g$) and yields a glassy thermoset with a high Young's modulus (~478 MPa)[39]. Thus, this single-vat g-DLP printing platform could achieve multi-property printing and integrate the soft organogel and glassy thermoset in a monolithic structure in one printing.

We select three different DoC states corresponding to three grayscale levels to showcase the multi-property cold-programmable SMP printing. These three materials are termed B1, B2, and B3, which are printed with increasing levels of grayscales (with a light intensity of 23.6 mW/cm² for B1 at 100% brightness, 15.4 mW/cm² for B2 at 80% brightness, and 3.1 mW/cm² for B3 at 40% brightness). Figure 1C displays the strain-stress behaviors obtained from the uniaxial tensile tests. B1 and B2 exhibit typical glassy and ductile behaviors, starting with a linear elastic deformation followed by yielding and subsequent strain softening up to more than 100% strain[41]. With continued loading, strain stiffening occurs due to polymer chain alignment. Both B1 and B2 can be stretched to more than 300% strain before failure at room temperature and even higher elastic stretchability at elevated temperature (Supplementary Fig. 1). The presence of a high molecular weight AUD crosslinker and the abundance of hydrogen bonds contribute to the ductile network, enabling large post-yielding deformation. This characteristic allows them to be cold-programmed in a glassy state without fracturing. Meanwhile, the low light intensity printed B3 is a soft and stretchable organogel. We further evaluate thermomechanical properties with dynamic mechanical analyses (DMA), as shown in Fig. 1D. The glassy B1 and B2 possess distinct $T_g$ of 87 °C and 66 °C, respectively, as determined from the peak of tan δ. The organogel, B3, has a much lower $T_g$ of −14 °C. The onset and offset of the two inflection points below and above $T_g$ divide the tanδ curve into three regions, corresponding to the glassy region, glass transition region, and rubbery region. For the convenience of discussions, we term the related temperature ranges as cold, warm, and hot zones. The onset temperature below $T_g$ is defined as $T_c$; for glassy SMPs, $T_c$ is higher than room temperature. SMPs programmed above $T_c$ in the warm or hot zone are classified as hot-programming, while those programmed below $T_c$ in the cold zone (generally at room temperature) are referred to as cold-programming. As marked in Fig. 1D, B1 and B2 have $T_c$ of 63 °C and 42 °C, respectively, both higher than room temperature.

We conduct hot- and cold-programming tests on glassy B1 to compare the shape memory behavior. Figure 1E shows the

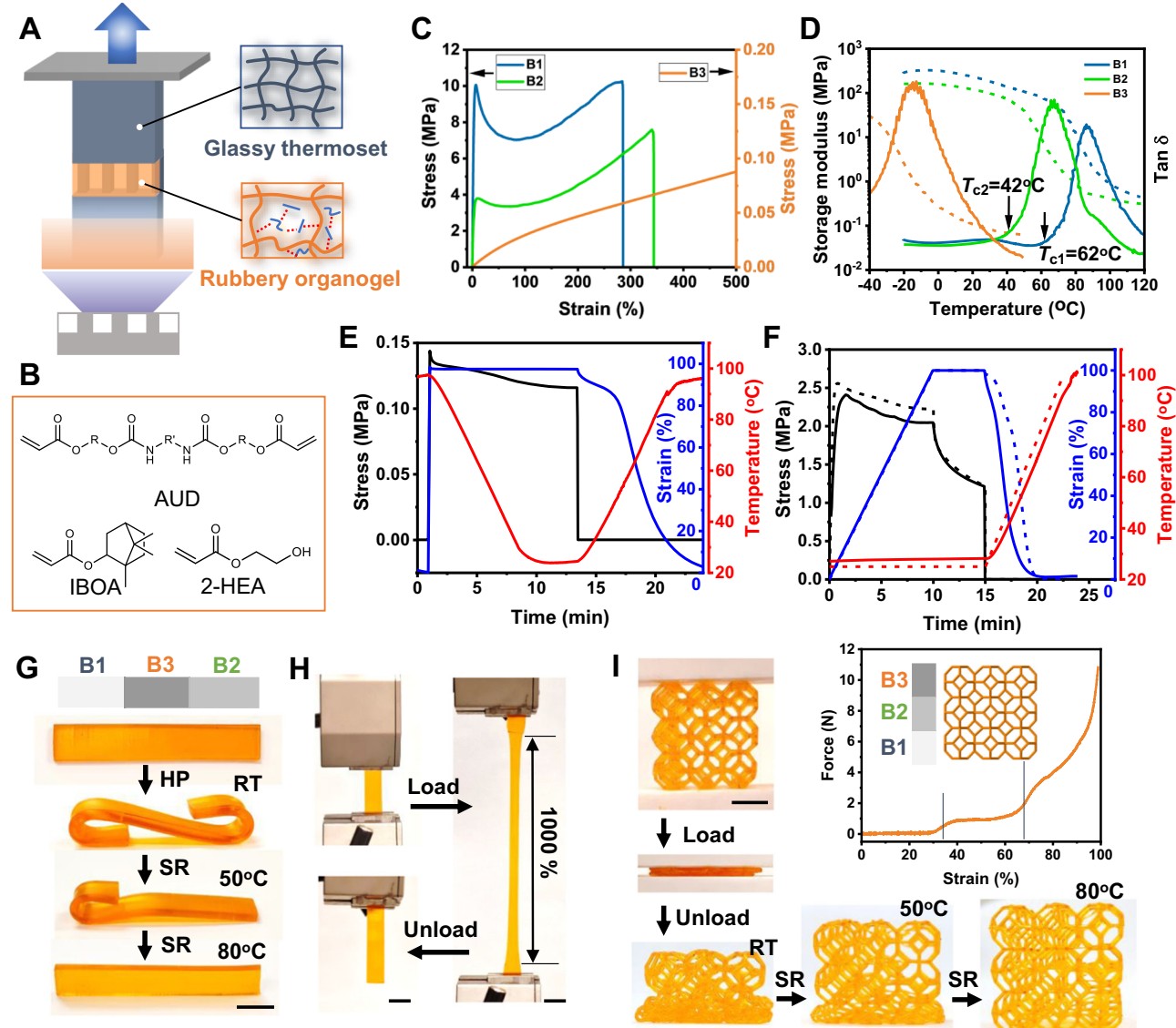

**Fig. 1 | Single-vat g-DLP multi-property 3D printing. A** Schematic illustration of the bottom-up g-DLP multi-property 3D printing. **B** Chemical structure of the monomers. **C** Stress-strain curves for B1, B2, and B3. **D** Storage modulus and tan δ as a function of temperature for B1, B2, and B3. The onset temperatures for B1 and B2 are marked as $T_{c1}$ and $T_{c2}$. **E** Hot-programming shape memory and recovery of B1. **F** Cold-programming shape memory and recovery of B1, the dash lines are results from the FEA simulation, and the solid lines are from the experiment. **G** A g-DLP printed strip with three grayscale levels and its hot-programming (HP) at 80 °C following shape recovery (SR) at 50 and 80 °C. **H** Elastic deformation of the middle region of B3. **I** Cold-compression programming of a graded lattice at room temperature (RT) and the corresponding three-stage force and strain curve, following shape recovery at 50 and 80 °C. All scale bar is 1 cm.

typical hot-programming shape memory behavior. SMP B1 is stretched with a programming strain $\varepsilon_P$ of 100% at the temperature of 100 °C (above $T_c$), and the programmed shape is fixed by decreasing the temperature to 25 °C (below $T_c$). The load is removed to obtain the fixed strain $\varepsilon_T$ and the shape fixity $r_T = \varepsilon_T/\varepsilon_P$ (96.9%). The temperature is then gradually increased for free recovery of the deformed sample, and the shape recovery ratio ($\varepsilon_T - \varepsilon_R)/\varepsilon_T$ (100%) is calculated using the strain after recovering $\varepsilon_R$. Figure 1F shows cold-programming shape memory by stretching. Similarly, SMP B1 is stretched by 100% at room temperature (below $T_c$) to obtain a temporary shape. The temperature is then increased without applying load to allow for free recovery. The printed SMP B1 shows an excellent shape fixing ratio of 95.8% and a shape recovery ratio of 99.4% even under large deformation, regardless of the programming temperature. Both the chemical crosslinking by AUD and hydrogen bonding generate netpoints within the thermoset network to stabilize the polyacrylate chain

segments and dictate the permanent shape of the network, contributing to such a high shape return capability. This work focuses on cold-programming of shape-morphing 4D printing. So, we further simulate the complete process of the cold-draw programming (i.e., stretching, holding, releasing) and heat-recovering of the glassy SMP B1 (Fig. 1F) and B2 (Supplementary Fig. 11) by using finite element analysis (FEA) (see "Methods" and Supplementary Note 2). The simulated stress and strain curves and the temperature history agree well with experimental results. A visual demonstration of the cold-draw programming and recovery of B1 and B2 is provided in Supplementary Fig. 2 and Supplementary Movie 1.

To prove the viability of g-DLP as a platform for shape-morphing 4D printing, two samples with graded properties are printed for verification. The first sample is a strip with two glassy ends (B1 and B2) and a rubbery middle section (B3), as shown in Fig. 1G. The sample is subjected to hot-programming at 80 °C to bend the glassy ends, and

then the programmed shape is fixed by rapid cooling in an ice bath. A two-stage recovery is tested by first increasing the temperature to 50 °C, which is above $T_{c2}$ (42 °C) but below $T_{c1}$ (62 °C). At this temperature, B2 gradually recovers to its original shape while B1 still preserves its temporary shape, shifting the global shape as shown in Fig. 1G. Upon further increasing the temperature to 80 °C above $T_{c1}$, the B1 end returns to its original shape, shifting the global shape back to its original state. Meanwhile, the rubbery middle section can be stretched by 1000% and recovers at room temperature (Fig. 1H). The second sample is a graded 3D lattice structure, as shown in Fig. 1I. It is subjected to cold-programming by compression at room temperature. The force-strain curve clearly shows three deformation regions corresponding to each layer during compression. Upon removing the load cell, only the rubbery B3 layer recovers. The compressed B2 and B3 layers recovers sequentially at temperatures of 50 °C and 80 °C. These experiments demonstrate the capability of g-DLP to fabricate SMPs with both hot and cold programming for shape-morphing 4D printing.

## Shape-morphing with cold-draw programming hinge

A classical mechanical actuated bilayer structure consists of an active layer and a passive one[42]. As discussed earlier, the glassy SMPs B1 and B2 both possess a relatively high $T_g$ above room temperature, which is critical to facilitate the fixing of the cold-programmed temporary shape. Once the stress is released, they exhibit unrecoverable deformation beyond the yield strain. This behavior is attributed to the presence of the glassy phase, which restricts the molecular mobility of the chain segments. Consequently, the stress becomes trapped as internal energy, causing the system to be locked into a non-equilibrium configuration. Meanwhile, the rubbery B3 instantly recovers at room temperature. Therefore, the glassy SMPs can be used for the active layer and rubbery B3 for the passive layer to produce a strain-induced bilayer bending structure upon a simple stretching at room temperature (we term this as cold-draw programming), as illustrated in Fig. 2A. The mismatch of strain recovery results in a bending deformation. However, it is difficult to cold-draw program the classical mechanical actuated bilayer structure because deforming the glass layer may require a large force. To overcome this issue, we employ the g-DLP platform to print a heterogeneous active composite hinge with glassy fibers (B1) embedded in a rubbery matrix (B3)[16], as shown in Fig. 2B. The optimized hinge design can be easily stretched at room temperature with low mechanical forces of around 2 N (Supplementary Fig. 3) while still retaining its glassy nature to preserve the bending deformation. The hinge structure is connected to glassy panels, and when a strain is applied, only the hinge module deforms, causing the connected parts to bend. When the temperature is above its $T_c$, the deformed fiber (SMP B1) returns to its original shape, and the hinge module also returns to its original state. This allows the hinge module to be programmed over many cycles.

It should be noted that in the case of single materials, the total applied programming stress can be divided into two portions: stress fixed and stress associated with springback[27]. The stress fixed is stored as internal energy, locking the polymer configuration and resulting in a fixed strain. On the other hand, the springback stress causes elastic/viscoelastic recovery upon unloading, which may take minutes to stabilize. However, in a multimaterial hinge, the rubbery matrix only undergoes elastic deformation, which helps in releasing the stress associated with springback and achieving faster stabilization of the hinge deformation. Meanwhile, the mechanical interaction of the two constituent materials can result in bending instead of uniaxial tension upon the mismatched strain recovery. As a result, to achieve fast and stable folding angles, the rational design of the multimaterial hinge structure based on mechanics principles is needed, and we thus develop analytical and FEA models for the hinge folding problem. More specifically, the folding angle of the hinge depends on the mechanical properties and geometric designs of two constituent

materials (i.e., hinge dimensions, fiber position, density, etc.). We derive an analytical model for the folding angle ($\theta$) of the hinge based on those parameters, and it can be expressed as

$$\theta = \kappa l_H (1 + \varepsilon_{NP}) \qquad (1)$$

where $\kappa$ is the bending curvature, $l_H$ is the length of the hinge and $\varepsilon_{NP}$ is the longitudinal strain (perpendicular to cross-section) of a neutral plane (detailed forms of relevant terms are given in Supplementary Note 3). Note that the use of Eq. (1) to predict $\theta$ requires the value of fixed strain $\varepsilon_T$, which depends on the programming strain $\varepsilon_p$ and the fixity $r_T$ governed by complex visco-plastic behavior of the glassy material (Supplementary Note 2.1). Here, an empirical formula of $r_T$ is obtained by fitting the experimental data, i.e., $r_T = -0.27\varepsilon_p^2 + 0.51\varepsilon_p + 0.71$, as shown in Fig. 2C. The larger programming strain $\varepsilon_p$ causes the reduced relaxation time and more viscous flow thus increasing the shape fixity, which agrees with previous report[36]. Using this empirical formula and Eq. (1), the theoretical prediction of the strain-angle relation achieves good agreement with FEA and experiment (Fig. 2C).

We verify the bending capability of the hinge module experimentally with the g-DLP printed samples and theoretically with FEA simulations, as shown in Fig. 2C, D. The g-DLP printing platform provides robust multi-property capabilities, enabling the heterogeneous integration of glassy and ductile fiber and stretchable matrix. This combination allows the hinge module to withstand large deformation and achieve a fully folded shape change (180° folding with a strain of 120%) of the hinge (Fig. 2D).

This understanding of the underlying mechanical principles provides a general design rationale for complex shape-morphing structures. By combining the cold-draw hinge modules, we are able to deform them in a controlled manner, enabling complex programmable shape-shifting. Since the hinge module bends towards the rubbery layer, this allows us to modify the bending direction of each hinge by printing the rubbery layer on the top side (for upward bending, white color hinges in the figures below) or the bottom side (for downward bending, orange color hinges). Figure 2E shows the g-DLP printed structures with alternated hinge modules on a stiff strip. The samples can be cold-draw programmed into an M or square shape with a predesigned bending direction. At the temperature above the $T_c$ of the fiber (i.e., 80 °C), it recovers to its original shape and allows for repeated programming. However, repeated large deformations may be accompanied by damage to the network, resulting in some irreversible deformation. This can impact the accuracy of shape morphing after multiple iterations of cycles. Therefore, the allowable deformation in complex structures should be carefully analyzed to ensure that the local deformation would not cause damage. In this case, the optimized hinge design undergoes 40 cycles of deformation and recovery, demonstrating good reversibility (Supplementary Fig. 4). Similarly, we can print more complex shape-morphing structures with multiple hinge modules to prescribe different configurations, as shown in Fig. 2F and Supplementary Movie 2. Moreover, the final configuration of the cold-draw shape-morphing structures is not only dictated by the prescribed hinge modules but also determined by the applied strain, providing additional reconfigurability after fabrication. Figure 2G and Supplementary Movie 3 further demonstrate this programmability, where a 2D strip with orientated hinge modules (30° along the long axis of the strip) induces out-of-plane bending of each module and results in a collective twist of the structure into a 3D helix shape. The pitch space can be controlled by varying the applied strain: a 60% applied strain led to a helix with a tighter pitch. The corresponding FEA simulations (Fig. 2F, G) based on the heterogeneous hinge structures are in good agreement with the experimental results, demonstrating the accurate angle and direction control of cold-draw shape morphing

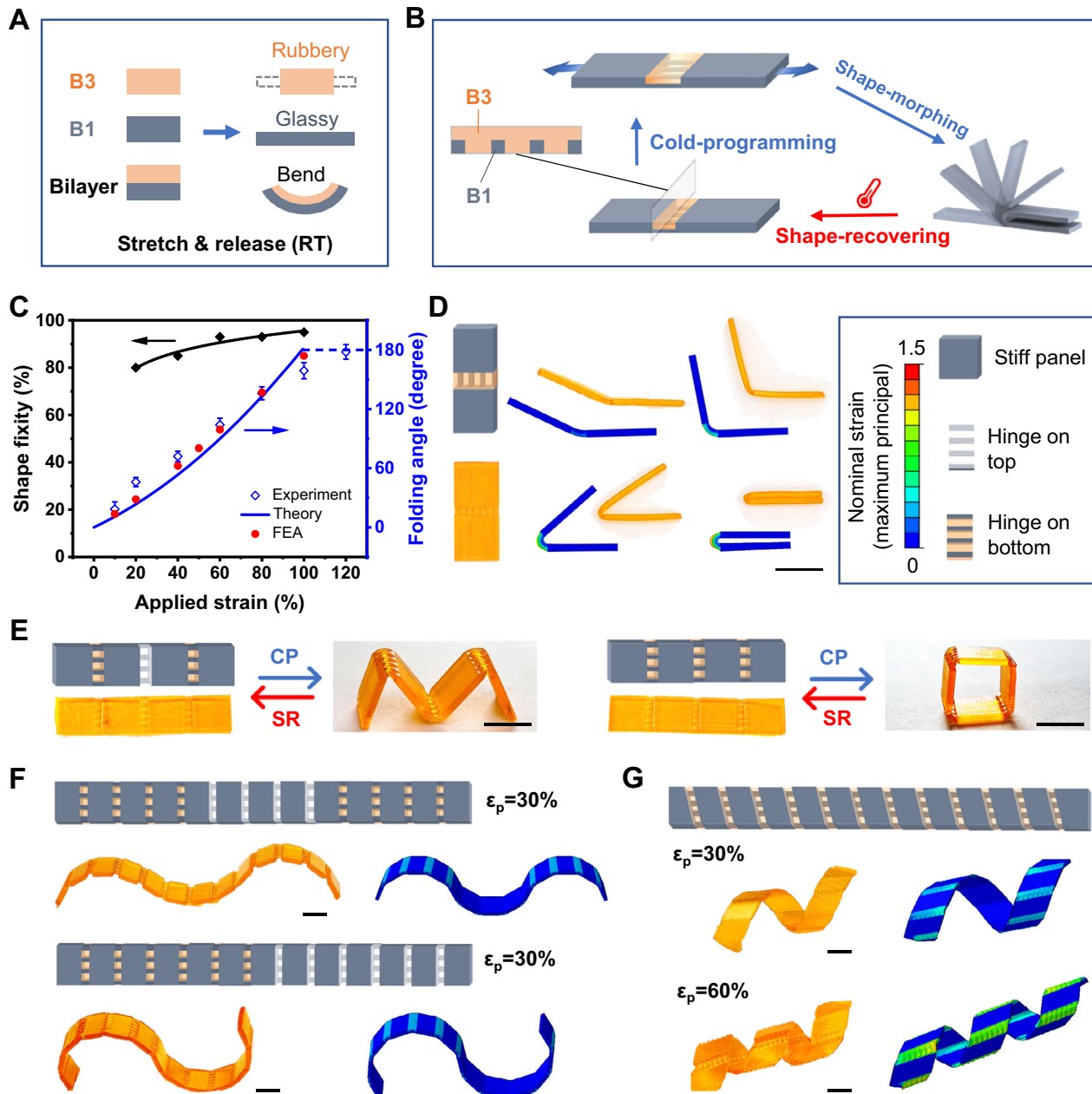

**Fig. 2 | g-DLP printed cold-programmable hinge. A** Mechanism of the bilayer bending. **B** Schematic of the heterogenous hinge module design and morphing. **C** Shape fixity of B1 under different strains and the experiment (error bars are standard deviations based on 5 experimental results), theory prediction, and FEA results of bending angles as the function of the applied strain of the hinge module. **D** Designed and printed hinge, and the experiment and FEA results of the cold-draw induced bending. **E** Cold-draw programming (CP) and shape-recovering (SR) of g-DLP printed structures. **F** Cold-draw programming with different hinge distributions. **G** Cold-draw programming of helices with 30%(upper) and 60%(lower) strain, respectively. The white color hinge indicates rubbery matrix on the top side, while the orange color hinge indicates rubbery matrix on the bottom side. (**F**) and (**G**) share the same color map as (**D**). All scale bar is 1 cm.

capability. Those shape-morphed structures can maintain their shape for months under room temperature without significant alterations.

**Programmable modular-controlled shape morphing structures**
As discussed above, unlike traditional hot-programming, which often requires heating the entire structure and thus limits the capability for local programming, cold-programming enables independent deformation of each hinge unit, resulting in a wide range of achievable configurations. To demonstrate the versatility of our approach, we design and print a human hand (Fig. 3A) using the hinge modules as joints with the hand palm and fingers printed in glassy B1. Due to room temperature cold-programming, each joint could be independently

programmed. Therefore, the printed hand is able to mimic different gestures, as seen in Fig. 3A and Supplementary Movie 4. This makes the printed hand a suitable gadget for human interaction.

Another example is demonstrated in Fig. 3B. The g-DLP printed 2D strip with an array of the hinges as the active units connecting with stiff panels (Fig. 3B). The circular hole on the panel can facilitate not only the precise controlling of the cold-draw programming strain using a universal material testing machine but also the assembly of through-hole interlocking structures in complex designs (as shown later). Each hinge could bend separately with a varied angle to achieve infinite shape configurability, as shown in Fig. 3B. Cold-programming shape morphing is an instant process that enables interactively programming

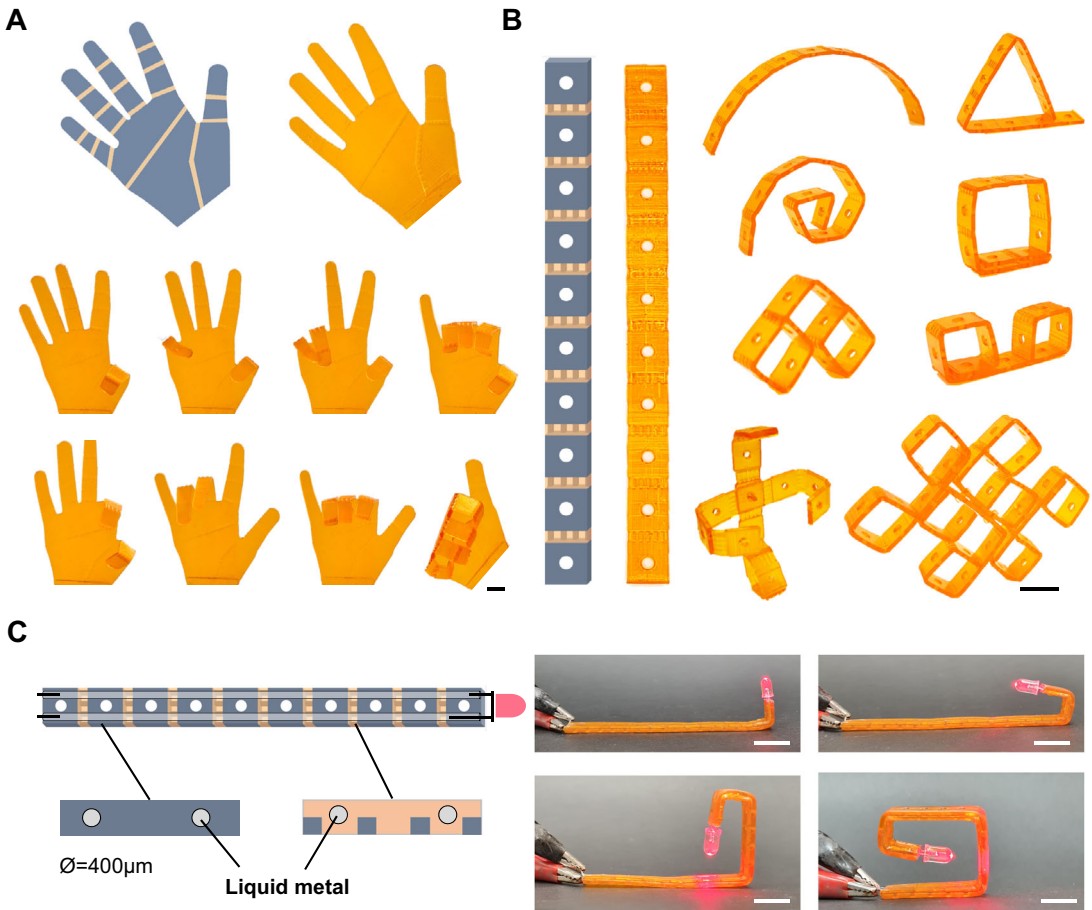

**Fig. 3 | Cold-programmed shape morphing with controllable local deformation. A** A g-DLP printed human hand with reconfigurability to mimic different gestures. **B** A strip with various configurations by localized regulating the hinge module and the through-hole interlocking assembly for complex 3D geometries. **C** Shape-morphing electronic device with conductive microchannel filled with liquid metal. All scale bar is 1 cm.

the shape with real-time feedback on the transformation and resulting geometries. The shape memory recoverability also allows rapid reconfiguration and iteration designs.

Shape morphing systems incorporated with conductors have emerged to create intelligent electronic devices with diverse adaptabilities, such as actuators, robots, antennas, etc.[43–45]. We utilize the high-resolution feature of g-DLP and add microchannels (diameter of 400 μm) to the strip design. The microchannels are then filled with liquid metal (eutectic gallium-indium (EGaIn)) by vacuum-assisted injection. The strip becomes a shape-morphing electronic device, as shown in Fig. 3C. The fabricated electronic strip can achieve variable configurations through smooth transition without interruption of the conductivity. This exemplifies the potential applicability of cold-draw shape morphing electronic devices through low-cost, high-resolution, and efficient g-DLP multi-materials 4D printing.

### Transformable panel structures

Rather than the one-direction stretchable shape-morphing strip, we further design a dual-direction stretchable planar sheet by connecting strips with vertical hinge modules, as shown in Fig. 4A. Three different prescribed 3D configurations are encoded in the 2D sheet through inverse design and can be achieved by cold-programming in specific directions. With a 50% stain in the $y$ direction, the 2D flat sheet folds into an open tube shape (Fig. 4B and Supplementary Movie 5). Applying the same strain in the $x$ direction results in the sheet rolling into a closed tube shape (Fig. 4C). Overlaying strains in both directions yields another different 3D shape (Fig. 4D). This process allows for preserving three encoded shapes in a single 2D flat sheet, greatly

increasing the programmability as compared to previous 4D printing methods.

Similarly, we print another flat sheet with all the hinge modules symmetrically placed in the sheet, as shown in Fig. 4E. Applying a 50% strain in any direction shifts it to the target configuration. Overlying strains in both directions result in a buckled-up 3D shape (Fig. 4E). The target configuration can be easily manipulated by changing the location of the hinge modules. As shown in Fig. 4F, flipping the hinges from the top side to the bottom changes the target shape from bulked-up to bulked-down. Our FEA simulations agree well with the experimental results, demonstrating precise control over the cold-draw programming activated shape morphing. More complex structures can be achieved by the through-hole assembly, as shown in Fig. 4G, further expanding the geometry adaptability. These transformable panel structures could potentially be used as substrates or structural matrices for fabricating shape-morphing antennas that can configure their radiation patterns for optimal performance[46].

### Multistage smart shape-morphing structures

The g-DLP printing platform offers a wide range of glass transition temperatures, and the selected materials from B1 to B3 exhibit different thermal-mechanical behaviors. As depicted in Fig. 5A, the $T_c$ for B1, B2, and B3 are schematically marked on the temperature strip to compare with room temperature and the selected programming temperature (50 °C). Below $T_c$, the polymer is in the cold zone (glassy region), which will retain the deformation. Above $T_c$, the polymer is in the warm/hot zone (glass transition/rubbery region), leading to full recovery of the deformation. As shown in Fig. 5B, both B1 and B2 have

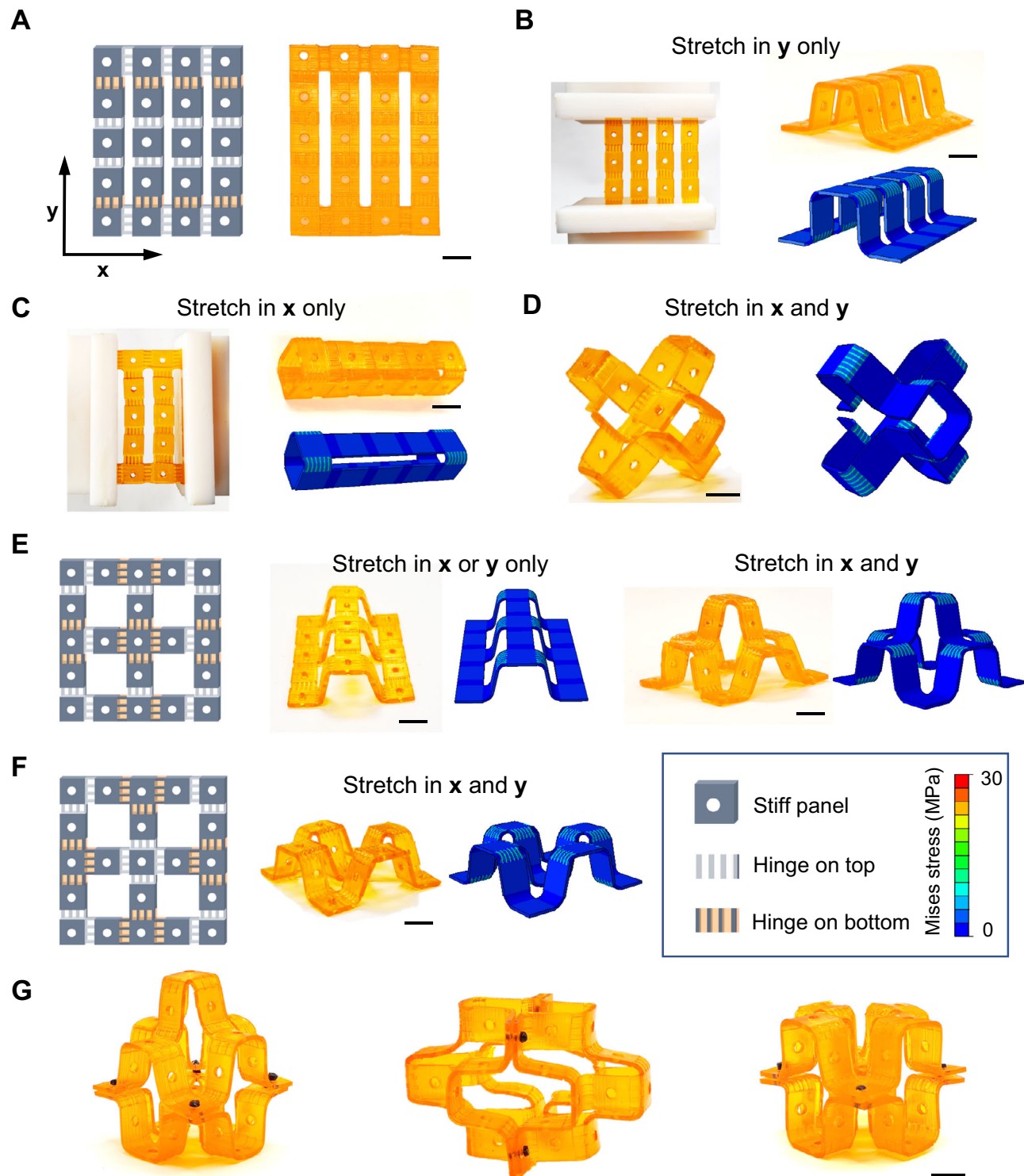

**Fig. 4 | Cold-programming transformable duel-direction panel structures.**
**A** Design and printed sample of a dual-direction shape-morphing structure.
**B**–**D** The configurations from experiments and FEA simulations with a 50% strain in different directions. **E**, **F** Design and printed sample of uni-direction shape morphing structures and the experiment and FEA results with a 50% strain on different directions. **G** through-hole interlocking assembly of complex 3D architectures with shape-morphing structures. All FEA simulations share the same color map. All scale bar is 1 cm.

higher $T_c$ than room temperature and can preserve the cold-programmed deformation. Increasing the temperature to 50 °C, B1 with a $T_c$ of 62 °C remains in its cold zone and retains the deformation, while B2 with a $T_c$ of 42 °C enters its warm zone (glass transition region), leading to a full recovery. The elastomer B3 has the lowest $T_c$ (below 0 °C) and immediately recovers upon the release of stretching

at room temperature. As such, we expect that hinge modules with different material combinations exhibit different responses to temperature. As shown in Fig. 5C, besides the initial hinge design (referred to as H13; 13 means the B1 and B3 are used), two additional designs using B2 to replace the rubbery matrix (H12) and glassy fiber (H23) are also demonstrated. Based on the previously described mechanism, the

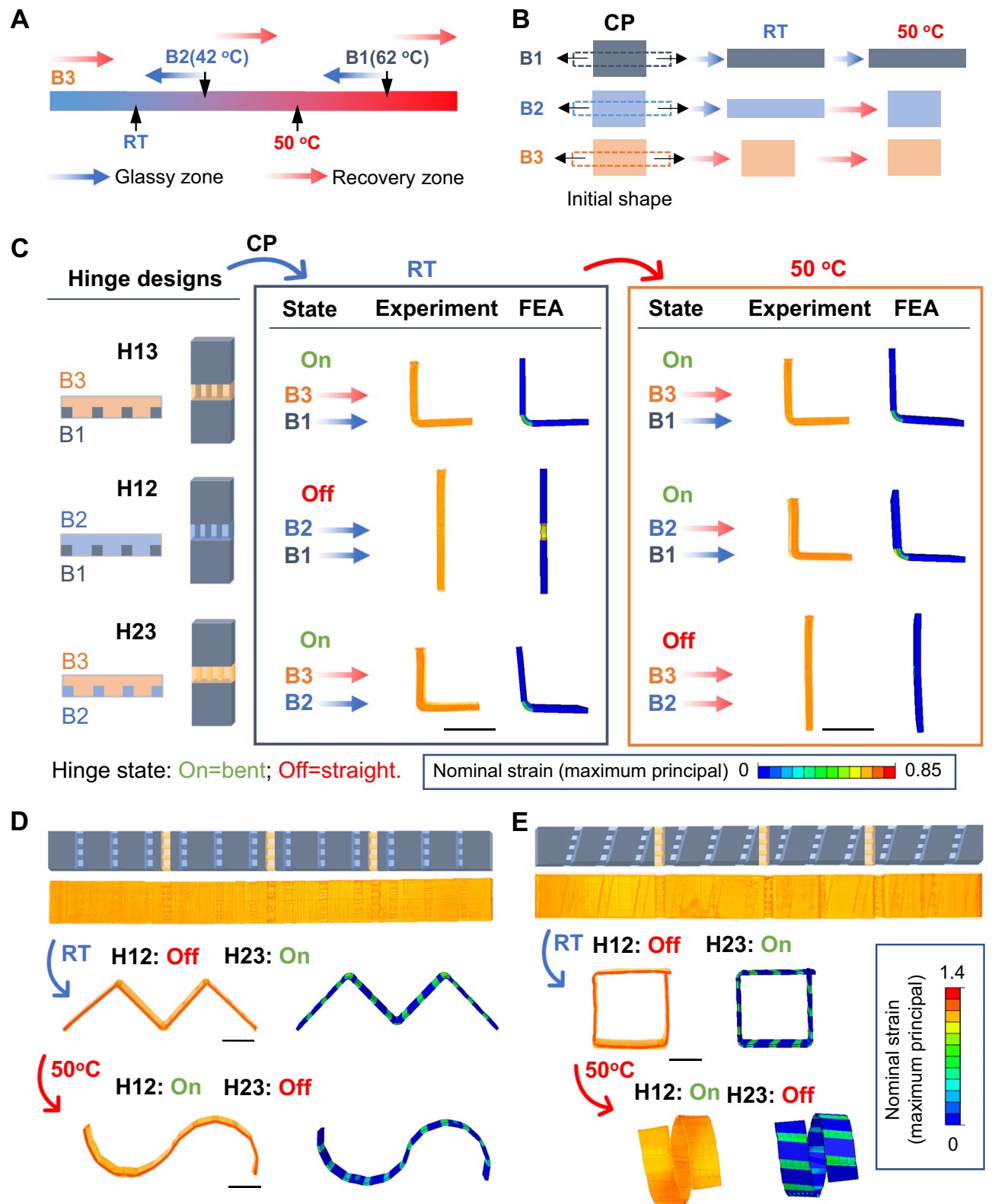

**Fig. 5 | Hinge design for two-stage shape morphing. A** Comparison of the onset temperature $T_c$ of B1 (62 °C), B2 (42 °C), and B3 (below 0 °C) with room temperature (RT) and the selected programming temperature (50 °C). Below $T_c$, the polymer is in the glassy zone (deformation is not recoverable), and above $T_c$ is in the recovery zone (deformation is recoverable). **B** Schematic of the temperature response of B1, B2, and B3 after stretching coldprogramming (CP) at room

temperature. **C** Design of the different hinge modules and the experiment and FEA results of their temperature-dependent response after stretching cold programming. The hinge is activated to the on state (bent) when the fiber and matrix materials are in a different zone and off state (straight) in the same zone. **D, E** Two-stage shape-morphing structures with hybrid hinge modules, and the experiment and FEA results in different temperatures. All scale bar is 1 cm.

activation of the hinge depends on the temperature-regulated activation states of the two constituent materials. Upon cold-programming, the hinge's state is determined by the following conditions: 1) if both materials are in the cold zone, the hinge will retain the strain without bending, and thus the hinge is off; 2) if one material is in the cold zone and the other is in the warm/hot zone, the hinge will bend due to the mismatch of the viscoelastic recovery and the hinge is on; 3) if both materials are in the warm/hot zone, the hinge will recover to the initial state, and the hinge is off.

We then experimentally demonstrate this concept. H13, at both room temperature and 50 °C, follows condition two and can be turned on (activated). H12 follows condition 1 at room temperature and is off; it follows condition 2 at 50 °C, and the hinge is on. H23 follows condition 2 at room temperature and the hinge is on; it follows condition 3 at 50 °C, and the hinge is off. Thus, the hinge module has been endowed with the smart ability to make autonomous decisions based on environmental temperatures.

Leveraging the differences between those thermo-responsive cold-programming hinges, we further design shape-morphing structures that can autonomously respond to environmental temperature changes and transit into pre-programmed configurations. Figure 5D showcases a g-DLP printed strip with hybrid hinge modules H12 and H23. The sample is subjected to cold-draw programming of all hinges. The H23 hinges are on instantly and turn the strip into an M shape, while H12 hinges are off. Increasing the temperature to 50 °C, H12 hinges become on, but H23 hinges recover, morphing the M shape into an S shape autonomously (Supplementary Movie 6). By switching the direction and location of the hinges, we can easily prescribe different multistage configurations, as exemplified in Fig. 5E. With the same morphing strategy, the strip is morphed into a square shape by activating H23 hinges at room temperature. The morphed in-plane square configuration can further progressively transform into an out-of-plane helix shape under changing temperatures by deactivating H23 and activating H12 (Supplementary Movie 6). These results demonstrate the feasibility of using g-DLP multi-property printing for the fabrication of cold-draw programmable smart shape-morphing structures. It also demonstrates the broad versatility of g-DLP for 3D/4D printing.

## Discussion

This work introduces the strategy of cold-programming for 4D printing of shape-morphing structures by using the single-vat-based multi-property g-DLP platform. The advanced feature of g-DLP is its ability to integrate different properties into a monolithic structure, greatly simplifying the fabrication of complex structures like the hinges, where stiff fibers are embedded in a rubbery matrix. The g-DLP platform leverages the combination of cold-programming and temperature-dependent shape-memory effects to create complex shape-morphing structures. The cold-programming enables instant and interactive shape-morphing, while the modular hinge design provides locally controllable deformation. The hinge structure can be easily adjusted to adapt to different structures and size scales (a submillimeter scale shape-morphing pipe is presented in Supplementary Fig. 5 and Supplementary Movie 7). The multi-property feature of g-DLP printing offers additional design options for smart multistage thermal responsive hinges. The proof-of-concept demonstration of smart shape-morphing structures with hybrid hinge designs shows the ability to create temperature-responsive, multistage morphing structures. Despite the advancements in g-DLP-based cold-programming 4D printing, there are areas that can be further improved. First, the rubbery matrix in the hinge module is an organogel with uncured photo monomers as the liquid phase, making it UV-sensitive. Consequently, exposure to UV light will deactivate the shape-morphable hinge structure. Applying necessary post-treatments, such as UV block coating or a secondary thermal reaction to consume the reactive groups, could be helpful[37]. Second, the cold-draw processing, with the

application of high strain, may result in some irreversible damage. This damage can affect the accuracy of deformation after multiple iterations of cycles. Optimizing the fiber density to ensure that the maximum required strain does not damage the network within the designed lifetime or introducing exchangeable bonding to restore the initial network when needed[47] can be beneficial. More functional g-DLP inks, beyond the one presented in this study, can be developed for various applications by following this strategy (an exemplary resin formulation is presented in Supplementary Fig. 6). Overall, the cold programming shape morphing strategy achieved with the g-DLP printing platform exhibits broad applicability for 4D printing. It brings potential advancement to the design and fabrication of morphing structures, including transformable metamaterial, shape-shifting antennas, reconfigurable electric devices, and many others.

## Methods

### Chemicals and materials

The photocurable resin is prepared by mixing 2-hydroxyethyl acrylate (technical grade, Sigma-Aldrich, MO, USA), isobornyl acrylate (technical grade, Sigma-Aldrich), and AUD (Ebecryl 8413, Allnex, GA, USA) with the weight ratio of 20:60:20. Then 1.2 wt% photoinitiator (Irgacure 819, assay 97%, Sigma-Aldrich) and 0.1 wt% photo absorber (Sudan I, assay 96%, Sigma-Aldrich) are added.

### g-DLP 3D printing

3D printing is performed with a bottom-up DLP printer that employed a 385 nm UV-LED light projector (PRO4500, Wintech Digital Systems Technology Corp., Carlsbad, CA, USA) and a linear translation stage (LTS150 Thorlabs, Newton, NJ, USA). A homemade container with an oxygen-permeable window (Teflon AF-2400, Biogeneral Inc., CA, USA) is used as the resin vat. The designed 3D structures are sliced into image files with a thickness of 0.05 mm and then converted into grayscaled image files with a MATLAB script. The continuous liquid interface production (CLIP) approach is utilized at the optimized speed of 3 s/layer to print the designed 3D structures (Supplementary Movie 8 depicts a typical printing process). The light intensity of the printer is calibrated with a photometer (ILT1400-A Radiometer, International Light Technologies Inc., MA, USA) before printing. The grayscale level is controlled by the brightness of the projector, with a light intensity of 23.6 mW/cm$^2$ for B1 at 100% brightness, 15.4 mW/cm$^2$ for B2 at 80% brightness, and 3.1 mW/cm$^2$ for B3 at 40% brightness.

### Finite element analysis (FEA) simulations

To model the stretching-releasing behavior of the hinge, full 3D FE simulations are performed using the commercial software ABAQUS (version 2018, Dassault Systèmes). The cold-programming and shape memory behavior of the glassy phase B1 and B2 are captured using a multi-branch viscoelastic model[36,48], which is implemented by combining a neo-Hookean model, a multi-branch model in the Prony series, and a temperature- and stress-dependent shift factor that specifies forms of time-temperature superposition principles and describes stress-induced shape memory effects. The rubbery phase B3 is assumed as an incompressible neo-Hookean solid as it is elastic in the studied temperature range (25–100 °C). Material parameters for materials with different grayscales are determined based on the uniaxial tension and DMA data. More details on the material models and parameters used in FEA simulations can be found in Supplementary Note 2. All the structures in simulations are in 3D and analyzed using C3D8H elements (eight-node linear brick, hybrid). The boundary conditions of the FE models are the same as those of the experiments. An implicit dynamics analysis with a quasi-static solver is adopted in the simulations to handle instabilities exhibited in the stretch-induced yielding and in the subsequent release shape morphing.

Detailed characterizations are provided in Supplementary Note 1.

## Data availability

The authors declare that the data supporting the findings of this study are available within the article and its Supplementary Information file. Raw data of the resin properties generated in this study are provided in the Source Data file with this paper. All other data is available from the corresponding author upon request. Source data are provided with this paper.

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

## Acknowledgements
H.J.Q. would like to acknowledge the support from an AFOSR grant (FA9550-20-1-0306; Dr. B.-L. "Les" Lee, Program Manager), Toyota North America, the gift funds from HP, Inc. and Northrop Grumman Corporation. This work is performed in part at the Georgia Tech Institute for Electronics and Nanotechnology, a member of the National Nanotechnology Coordinated Infrastructure, which is supported by the National Science Foundation (ECCS-1542174).

## Author contributions
L.Yue and H.J.Q. conceived the concept. L.Yue, X.S., Y.S., M.T., and T.N. designed the experiments. X.S. performed the FEA simulations. L.Yu, X.S. and S.M.M. conducted the structures design. L.Yu and M.L. performed the property characterization. L.Yue and H.J.Q. prepared the manuscript. All authors discussed the results and commented on the paper.

## Competing interests
L.Yue, H.J.Q., Y.S., and M.T. are coinventors on a provisional patent application. The remaining authors declare that they have no competing interests.
