## [Peer Review File · Nature Communications]

Cold programmed shape morphing structures based on
grayscale digital light processing 4D printingEditorial Note: Parts of this Peer Review File have been redacted as indicated to remove third-party material where no permission to publish could be obtained.

REVIEWER COMMENTS

Reviewer #1 (Remarks to the Author):

The manuscript titled “Cold programmed shape morphing structures based on grayscale digital light processing 4D printing” prepared shape-morphing structures with cold programming capabilities using grayscale digital light processing (g-DLP) based 3D printing. However, the same material and preparation method has been reported by the authors previously. Although the authors explore new properties of shape memory and cold programming capabilities, they failed to clarify their mechanisms. The authors should clarify the novelty and advancement of this work more clearly, since the cold drawing is a conventional polymer manufacturing process. Therefore, I do not recommend the publication of the manuscript in Nature Communications:

1. The authors reported a kind of shape morphing structures based on grayscale digital light processing 4D printing. What is the solution for printing?
2. The authors have previously reported the same material, prepared using the same method, and demonstrated its elasticity (Nature Communications, (2023) 14:1251, Figure 2). However, the present work focuses on the material's shape-memory properties, and therefore, the authors should provide an explanation of the underlying mechanism behind this behavior. What kind of molecular interaction fixed the temporary shape during the cold-programing process. Additionally, it would be helpful for the authors to clarify why this material can undergo cold drawing.
3. The term "cold-programmable 4D printing" may be unclear. In this study, the authors utilized grayscale digital light processing 3D printing to prepare the materials, followed by a cold drawing process. However, the relationship between these two steps is not entirely clear. The authors should clarify whether the 3D printing process facilitates the cold-drawing step, particularly given their claim that cold-drawing the shape-memory polymer is typically challenging.
4. The author claims that their materials have shape recovery ratios higher than 90%. To

provide more direct evidence of this phenomenon, the authors should consider supplementing their work with movies or photos showing the materials recovering their permanent shape. For instance, a 100% cold-drawn film that can shrink to its original length when heated would be a suitable demonstration. Although Figure 1G is a clear demonstration, it is prepared by hot programming. Figure 1H presents the cold compression and recovering process, but the structure is too complicated to directly observe the materials recovery performances. It would be beneficial to supplement the data with direct visual evidence of shape recovery after cold programming.

5. For cold-programmed shape morphing structures, what is the reversibility? Does this structure can morph normally after 100 cycles or more?

6. The multi-stage smart shape-morphing structures described in Figure 5 is quite interesting. As the author discussed, the selected programming temperature is 50 °C. How accurately this structure can respond to the temperature change, such as 45 °C, 50 °C, and 55 °C? Is it possible to set more than one programming temperature (50 °C, 60 °C, 70 °C, etc.), so that the shape-morphing structures can have more morphing abilities?

7. The examples in the paper are in centimeter scale. Is it possible to print shape-morphing structures in millimeter or sub-millimeter scale? This would enable more applications.

Reviewer #2 (Remarks to the Author):

I read with great interest the manuscript by Qi's group which describes an interesting way of using gray-scale DLP printing to yield complex 3D temporary shapes by simple cold-stretching (or cold programming). The printing method and material formulation have been previously reported by the authors. The innovation in the current work is the mechanical design of the multi-material distribution to access various designable temporary shapes that can recover upon heating. The work, in terms of the mechanical hinge design using multi-materials, takes morphing hinges to a different level. For this reason, I support its publication by Nature Communications with the following comments for the authors to consider for revision.

1. Cold-programming of SMP has been widely known, but never actually caught major attention. This is probably due to the strong dependence of shape fixity on the strain, making it uneasy to control. This aspect is investigated here for the individual single material

formulation. I would encourage the authors to discuss this in more depth on how this impacts representative multi-material hinges, with either simulation or experimental results.

2. Authors are also encouraged to discuss limitations and/or remaining challenges of the current approach in the discussion section.

3. Other methods to produce complexed shape-morphing hinges consisting of multi-materials have been reported (e.g. Sci. Adv. 2020, 6, eaaz2362), although they are not related to cold programming of 3D printed SMP. Morphing hinges have also been reported by other 4D printing methods. Authors are encouraged to expand the introduction so that potential readers can gain a broader overview of the topic.

Reviewer #3 (Remarks to the Author):

In this work, the authors present a new 3D printing strategy for reprogrammable shape morphing structures under cold programming. The 3D printing is based on the authors' grayscale digital light processing (g-DLP) method. The mechanical characterizations of the printed materials show outstanding properties in terms of high elongation strain and high toughness, which are key factors for programming shape morphing structures under large deformation. The authors demonstrate the capability of such material and the 3D printing method by designing different hinges for programmable 3D structures that are formed from 2D patterns. The morphing is enabled by stretching the printed material at room temperature. The reported method is a promising new strategy for advanced and easy manufacturing of complicated 3D structures.

I believe this work is timely and of great interest to the field of 3D printing and general polymer physics. The paper is well-written, and the scientific insights are clearly discussed. I recommend the publishing of this paper without delay. Here are a couple of minor comments and questions.

- The reported material showed outstanding stretchability before failure and good shape recovery performance. The reviewer is curious about the maximum elastic strain at different temperatures. Can the authors provide this information?
- For the shape morphing demonstrations enabled by the mismatch strain, do the shapes

change their configurations with time, i.e., due to stress relaxation of the viscous material?

How long can the shapes be kept?

- Typo in Figure 2c: Fixicity.
- In Figure 2D, why is the nominal strain at the compressive portion of the hinge zero? It should be negative strain. The authors should double check or clarify. Do the FEA simulations in F and G share the same strain color bar as D?

Responses to Reviewers

Dear reviewers,

Thank you for reviewing of our manuscript. We appreciate your feedback and comments on our manuscript. Based on your insightful suggestion and comments, we have carefully revised our manuscript. Please see below for a point-by-point response to the comments and associated modifications to the manuscript.

Reviewer #1

General comments:

The manuscript titled “Cold programmed shape morphing structures based on grayscale digital light processing 4D printing” prepared shape-morphing structures with cold programming capabilities using grayscale digital light processing (g-DLP) based 3D printing. However, the same material and preparation method has been reported by the authors previously. Although the authors explore new properties of shape memory and cold programming capabilities, they failed to clarify their mechanisms. The authors should clarify the novelty and advancement of this work more clearly, since the cold drawing is a conventional polymer manufacturing process. Therefore, I do not recommend the publication of the manuscript in Nature Communications:

Response: We appreciate the reviewer for dedicating the time to thoroughly review our manuscript and for providing constructive comments and suggestions. The significance of our previous work on multi-material g-DLP paper lies in its resin design strategy, which can replicate the high elastic stretchability of organogel at a moderate conversion while achieving the high stiffness of thermosets at high conversion. This strategy serves as a universal approach for g-DLP resin design. However, in this study, we are implementing the g-DLP platform for shape-morphable 4D printing. We believe that this methodology offers an advanced and simplified manufacturing approach for creating complex shape-morphable devices, as suggested by Reviewer 2 and Reviewer 3. It is not necessary to use the exact same ink as reported in our previous paper (as illustrated in Comment #2). It is possible to follow this strategy to develop alternative functional inks for more specific

applications. We hope that our work will inspire the academic community working in this field to explore new possibilities.

Below, we provide a point-by-point response.

Comments #1: The authors reported a kind of shape morphing structures based on grayscale digital light processing 4D printing. What is the solution for printing?

Response: The shape-morphing structures based on g-DLP 4D printing utilize the integrated multi-material features in a single-vat, one-step printing process. This represents a significant advancement achieved through the solution we developed in our previous publication (Nat Commun 14, 1251, 2023). In our resin design strategy, as described in the paper, the light intensity of each pixel is adjusted during printing based on the assigned properties. Higher light intensity results in a stiff thermoset, while lower light intensity produces a soft and stretchable organogel. The difference in network mechanisms is illustrated in Figure R1. The abundance of hydrogen bonding within the monomers and the gelled network contributes to the stretchable nature of the organogel. The resolution of printing depends on the pixel size, which typically ranges between 20 μm and 100 μm . In this work, the pixel size is 50 μm . As demonstrated in our response to comment #7, we were about to print fibers that are 100 μm by 100 μm , indicating the capability to achieve high resolution.

Figure R1. Depiction of the chemical makeup of the g-DLP ink used to obtain orders of magnitude stiffness difference.

Comments #2: The authors have previously reported the same material, prepared using

the same method, and demonstrated its elasticity (Nature Communications, (2023) 14:1251, Figure 2). However, the present work focuses on the material's shape-memory properties, and therefore, the authors should provide an explanation of the underlying mechanism behind this behavior. What kind of molecular interaction fixed the temporary shape during the cold-programming process. Additionally, it would be helpful for the authors to clarify why this material can undergo cold drawing.

Response: We appreciate the reviewer for raising this excellent point. The mechanism behind the temporary shape fixity in cold draw programming is attributed to the strong constraints imposed on the polymer chains and segments during the large deformation above the yielding point. Here, the glassy phase of stiff polymers restricts the molecular mobility of their chain segments. Cold drawing (or large deformation) can help the molecular chain overcome the energy barrier and lead to stress-induced molecular movement. Upon removing the external force, the molecules stay in the new state, showing plastic-like deformation, which effectively traps the stress in the form of internal energy and locks the system into a non-equilibrium configuration (J. Polym. Sci. Part B: Polym. Phys., 54,1319-1339,2016)

In response, we have added the following discussion in the Shape-morphing with cold-draw programming hinge section:

“As discussed earlier, the glassy SMPs B1 and B2 can be cold programmed at room temperature. Once the stress is released, they exhibit unrecoverable deformation beyond the yield strain. This behavior is attributed to the presence of the glassy phase, which restricts the molecular mobility of the chain segments. Consequently, the stress becomes trapped as internal energy, causing the system to be locked into a non-equilibrium configuration.”

Regarding why the material can undergo cold drawing, it requires deformation without damage beyond the yielding point. Not all SMPs are capable of cold drawing programming; in fact, many thermoset SMPs tend to be brittle in their glassy state and may fracture before yielding or right after yielding. In our case, the presence of a high molecular weight AUD crosslinker and the formation of hydrogen bonds result in a ductile network, as illustrated

in Figure 1c. These two factors facilitate the capability of large post-yielding deformation.

Following the reviewer's comment, we have revised the related content in the Single vat g-DLP printing of materials with multi-property section:

“Both B1 and B2 can be stretched to more than 300% strain before failure at room temperature, and even higher stretchability at elevated temperature. The presence of a high molecular weight AUD crosslinker and the abundance of hydrogen bonds contribute to the ductile network, enabling large post-yielding deformation. This characteristic allows them to be cold-programmed in a glassy state without fracturing.”

The above described strategy is not limited to specific materials. By employing the same ink design strategy, we have formulated an alternative ink (2-[[*N*-(Butylamino)carbonyl]oxy]ethyl acrylate (Sigma-Aldrich, MO, USA), acrylate acid (Sigma-Aldrich, MO, USA) and AUD (Ebecryl 8413, Allnex, GA, USA) were mixed with the weight ratio of 2:2:1, Figure R2a) for demonstration purposes. We printed the same hinge structure using this ink, and it exhibited the same capability for cold programming. We believe that this methodology represents a general approach for 4D printing.

In response, we added following sentence in discussion section:

“More functional g-DLP inks, beyond the one presented in this study, can be developed for various applications by following this strategy (an exemplary resin formulation is presented in Supplementary Figure 6).”

We hope that the publication of this work in Nature Communications will inspire the research community to develop more functional g-DLP inks, such as electroconductive materials, for creating shape-morphing structures based on our methodology.

Figure R2 (added as Supplementary Fig. 6). A) Formulation of an alternative g-DLP ink; strain-stress curves of g-DLP printed with the alternative resin of B) stiff thermoset state printed with a light intensity of 23.6 mW/cm² and C) rubbery organogel state printed with a light intensity of 3.1 mW/cm²; D) cold draw performance of the printed hinge structure.

Comments #3: The term "cold-programmable 4D printing" may be unclear. In this study, the authors utilized grayscale digital light processing 3D printing to prepare the materials, followed by a cold drawing process. However, the relationship between these two steps is not entirely clear. The authors should clarify whether the 3D printing process facilitates the cold-drawing step, particularly given their claim that cold-drawing the shape-memory polymer is typically challenging.

Response: We appreciate the reviewer for raising this important point. Cold drawing of SMPs is typically challenging. It requires the polymer to be ductile at room temperature. On the other hand, achieving good shape fixity necessitates the polymer to be glassy and stiff, which makes it difficult to draw. The g-DLP platform we developed, along with the ink design strategy, serves as a valuable tool for fabricating variable shape-morphing structures, as demonstrated in the manuscript. Meanwhile, cold-drawing or the use of water/oven bath serves as the triggering for shape morphing. The advanced feature of g-

DLP is its ability to integrate different properties into a monolithic structure, greatly simplifying the fabrication of complex structures like the hinge structure presented in our work, where stiff fibers are embedded in a rubbery matrix. The triggering force and temperature can be easily adjusted by varying the fiber density and properties. However, these features rely on the significant contrast in properties between the materials. The ink polymer used in g-DLP was specifically designed for this purpose. Thus, with this platform, we can easily adjust the triggering force and temperature of the hinge module. As shown in Figure S1, the hinge structure can be easily hand-drawn. For a similar multi-material-based shape morphing structure, whether it is force-triggered or temperature-triggered, it would typically require a more complicated fabrication process rather than this presented g-DLP printing.

Comments #4: The author claims that their materials have shape recovery ratios higher than 90%. To provide more direct evidence of this phenomenon, the authors should consider supplementing their work with movies or photos showing the materials recovering their permanent shape. For instance, a 100% cold-drawn film that can shrink to its original length when heated would be a suitable demonstration. Although Figure 1G is a clear demonstration, it is prepared by hot programming. Figure 1H presents the cold compression and recovering process, but the structure is too complicated to directly observe the materials recovery performances. It would be beneficial to supplement the data with direct visual evidence of shape recovery after cold programming.

Response: We thank the reviewer for the valuable suggestion. In response to the reviewer's recommendation, we have included the following demonstration. The printed samples B1 and B2 were cold-drawn programmed. The cold-drawn deformation was well maintained at room temperature. At 50°C, B2 successfully recovered its original shape, while B1 remained in the deformed state. At 80°C, B1 also fully recovered its original shape. We have added a supplementary movie (Supplementary Movie 1) showcasing this demonstration.

Figure R3 (added as Supplementary Fig. 2). A) B1 and B2 sample as printed; B) B1 and B2 sample after cold-drawing processing; C) B1 and B2 sample after 50 °C heat treatment, only B2 recovered; D) after 80 °C heat treatment, B1 recovered. Video was provided as Supplementary Movie 1.

Comments #5: For cold-programmed shape morphing structures, what is the reversibility? Does this structure can morph normally after 100 cycles or more?

Response: This is a great question. As in any polymers, if the applied strain is too large, the material may damage and cause some irreversible deformation. This damage can impact the accuracy of deformation after multiple iterations of cycles. Therefore, the allowable deformation in complex structures should be carefully analyzed to ensure that the local deformation would not cause damage. In our case, we performed the H13 hinge manually for 40 cycles deformation-recovery. The hinge performance remains constant.

Figure R4 (added as Supplementary Fig. 4). H13 hinge cold-drawing performance for various cycles and recover to initial state after 40 cycles.

In response to the potential issue, we have revised the Shape-morphing with cold-draw programming hinge section to address:

“At the temperature above the T_c of the fiber (i.e., 80°C), it recovers to its original shape and allows for repeated programming. However, repeated large deformations may be accompanied by damage to the network, resulting in some irreversible deformation. This can impact the accuracy of shape morphing after multiple iterations of cycles. Therefore, the allowable deformation in complex structures should be carefully analyzed to ensure that the local deformation would not cause damage. In this case, the optimized hinge design underwent 40 cycles of deformation and recovery, demonstrating good reversibility (Supplementary Fig. 4).”

And added following content in discussion section:

“Secondly, the cold draw processing, with the application of high strain, may result in some irreversible damage. This damage can affect the accuracy of deformation after multiple iterations of cycles. Optimizing the fiber density to ensure that the maximum required strain does not damage the network within the designed lifetime or introducing exchangeable bonding to restore the initial network when needed,⁴⁸ can be beneficial.”

Comments #6: The multi-stage smart shape-morphing structures described in Figure 5 is quite interesting. As the author discussed, the selected programming temperature is 50 °C. How accurately this structure can respond to the temperature change, such as 45 °C, 50 °C, and 55 °C? Is it possible to set more than one programming temperature (50 °C, 60 °C, 70 °C, etc.), so that the shape-morphing structures can have more morphing abilities?

Response: This is an interesting question. In the reviewing paper by Li et al. (Ref. 27), the thermal mechanical behavior of shape memory polymers is divided into three regions: cold, warm, and hot. These regions correspond to the glassy region, glass transition region, and rubber elasticity region, respectively. For a deformed SMP, recovery occurs within the

warm and hot zones. In the manuscript, the critical temperature at the onset point of $\tan \delta$ is defined as T_c . Below T_c , the polymer is in a glassy state, while above T_c , the polymer exhibits viscoelastic or rubbery behavior. The T_c values for B1 and B2 are 62°C and 42°C , respectively. Any temperature within this range (42 to 62°C) can trigger the designed shape shifting.

To ensure a distinct response to the set temperature, it is important to maintain a relatively large difference in T_c (such as 20°C in this work). For more stages of temperature-responsive hinges, a material with a high T_c , such as 80°C , would be required. This would correspond to a glass transition temperature of around 110°C or higher. It is possible to design a g-DLP ink with a higher maximum T_c , which would enable additional morphing temperatures.

[FIGURE R5A REDACTED]

Figure R5. A) Schematic of $\tan \delta$ as a function of temperature to demonstrate the “cold,” “warm,” and “hot” zones during programming (Figure 1 from Ref. 27); B) $\tan \delta$ curves of the printed samples and the related T_c for B1 and B2 (Figure 1D in manuscript).

Comments #7: The examples in the paper are in centimeter scale. Is it possible to print shape-morphing structures in millimeter or sub-millimeter scale? This would enable more applications.

Response: We appreciate the reviewer for this valuable comment. The resolution of g-DLP printing relies on the pixel size of the projector. The printing setup we are utilizing has a pixel size of $50\ \mu\text{m}$. In response to the reviewer's feedback, we have designed a sub-

millimeter scale pipe with an inner diameter of $500\ \mu\text{m}$ and a fiber size of $100*100\ \mu\text{m}$ embedded within the rubbery matrix. The performance of this design is showcased in Supplementary Movie 7.

Figure R6 (added as Supplementary Fig. 5). A) Design of hinge structure in a shape morphable pipe; B) the cross section of the printed sample; C) shape changing of the printed pipe after cold-draw programming; D) temperature response of the pipe performed in 25 and 80°C. Experimental video was provided as Supplementary Movie 7.

In response, we have added the following sentence in discussion section:

“The hinge structure can be easily adjusted to adapt to different structures and size scales (a sub-millimeter scale shape-morphing pipe was presented in Supplementary Fig. 5 and Supplementary movie 7).”

Reviewer #2

General comments:

I read with great interest the manuscript by Qi's group which describes an interesting way of using gray-scale DLP printing to yield complex 3D temporary shapes by simple cold-stretching (or cold programming). The printing method and material formulation have been previously reported by the authors. The innovation in the current work is the mechanical design of the multi-material distribution to access various designable temporary shapes that can recover upon heating. The work, in terms of the mechanical hinge design using multi-materials, takes morphing hinges to a different level. For this reason, I support its publication by Nature Communications with the following comments for the authors to consider for revision.

Response: We appreciate the reviewer for recognizing the novelty of our work and recommending its publication in Nature Communications. In the following, we have addressed each specific comment in our point-by-point responses.

Comments #1: Cold-programming of SMP has been widely known, but never actually caught major attention. This is probably due to the strong dependence of shape fixity on the strain, making it uneasy to control. This aspect is investigated here for the individual single material formulation. I would encourage the authors to discuss this in more depth on how this impacts representative multi-material hinges, with either simulation or experimental results.

Response: We appreciate the reviewer for this valuable comment. We agree with the reviewer that cold drawing of SMPs is typically challenging. It requires the polymer to be ductile at room temperature. On the other hand, achieving good shape fixity necessitates the polymer to be glassy and stiff, which makes it difficult to draw. The g-DLP platform we developed, along with the ink design strategy, simplified the fabrication of multi-material hinges with the required properties. To ensure shape fixity, we selected a glassy state B1 polymer with a Tg of 87°C as the fiber component in the representative H13 hinge. In the case of single materials, the total programming stress applied can be divided into stress fixed and stress associated with springback. The stress fixed is stored as internal

energy, locking the polymer configuration, and resulting in fixed strain. The springback stress causes elastic/viscoelastic recovery upon unloading, which may take minutes to stabilize. However, in a multi-material hinge, the rubbery matrix only undergoes elastic deformation. The recovery force of the rubbery matrix and the springback-associated force of the glassy can be easily adjusted by varying the fiber density. Additionally, the recovery time of the rubbery matrix is significantly shorter than the viscoelastic springback, leading to faster stabilization and improved shape fixity in multi-material hinges compared to single materials. We have provided simulations and experimental results in Figure 2c as well as detailed calculation the Supporting Information.

Following the reviewer's comment, we have added the following discussion the Shape-morphing with cold-draw programming hinge section:

“It should be noted that in the case of single materials, the total applied programming stress can be divided into two portions: stress fixed and stress associated with springback.²⁷ The stress fixed is stored as internal energy, locking the polymer configuration, and resulting in a fixed strain. On the other hand, the springback stress causes elastic/viscoelastic recovery upon unloading, which may take minutes to stabilize. However, in a multi-material hinge, the rubbery matrix only undergoes elastic deformation, which helps in releasing the stress associated with springback and achieving faster stabilization of the hinge deformation. Meanwhile, the mechanical interaction of the two constituent materials can result in bending instead of uniaxial tension upon the mismatched strain recovery. As a result, to achieve fast and stable folding angles, the rational design of the multi-material hinge structure based on mechanics principles is needed, and we thus developed analytical and FEA models for the hinge folding problem.”

Comments #2: Authors are also encouraged to discuss limitations and/or remaining challenges of the current approach in the discussion section.

Response: We thank the reviewer for this valuable suggestion. There are two potential challenges that may affect the performance of the multi-material hinge structure. Firstly, the rubbery matrix in the hinge module is an organogel with uncured photo monomers as the liquid phase, making it UV sensitive. Consequently, exposure to UV light will

deactivate the shape-morphable hinge structure. Applying necessary post-treatments, such as UV block coating or a secondary thermal reaction to consume the reactive groups, could be helpful. Secondly, the cold draw processing, with the application of high strain, may result in some irreversible damage. This damage can affect the accuracy of deformation after multiple iterations of cycles. Optimizing the fiber density to ensure that the maximum required strain does not damage the network within the designed lifetime or introducing exchangeable bonding to restore the initial network when needed (Xie et al., Nat Commun 14, 1313 ,2023) can be beneficial.

In response, we have revised the manuscript in the discussion section to offer a more comprehensive perspective on this approach as the reviewer suggested.

“Despite the advancements in g-DLP-based cold programming 4D printing, there are areas that can be further improved. Firstly, the rubbery matrix in the hinge module is an organogel with uncured photo monomers as the liquid phase, making it UV sensitive. Consequently, exposure to UV light will deactivate the shape-morphable hinge structure. Applying necessary post-treatments, such as UV block coating or a secondary thermal reaction to consume the reactive groups, could be helpful.³⁷ Secondly, the cold draw processing, with the application of high strain, may result in some irreversible damage. This damage can affect the accuracy of deformation after multiple iterations of cycles. Optimizing the fiber density to ensure that the maximum required strain does not damage the network within the designed lifetime or introducing exchangeable bonding to restore the initial network when needed,⁴⁸ can be beneficial.”

Comments #3: Other methods to produce complexed shape-morphing hinges consisting of multi-materials have been reported (e.g. Sci. Adv. 2020, 6, eaaz2362), although they are not related to cold programming of 3D printed SMP. Morphing hinges have also been reported by other 4D printing methods. Authors are encouraged to expand the introduction so that potential readers can gain a broader overview of the topic.

Response: We would like to thank the reviewer for bringing up this peering working in 4D printing shape morphing structures. In response, we have added Sci. Adv. 2020, 6, eaaz2362 as Ref. 19 in the revised manuscript to acknowledge this work.

Reviewer #3

General comments:

In this work, the authors present a new 3D printing strategy for reprogrammable shape morphing structures under cold programming. The 3D printing is based on the authors' grayscale digital light processing (g-DLP) method. The mechanical characterizations of the printed materials show outstanding properties in terms of high elongation strain and high toughness, which are key factors for programming shape morphing structures under large deformation. The authors demonstrate the capability of such material and the 3D printing method by designing different hinges for programmable 3D structures that are formed from 2D patterns. The morphing is enabled by stretching the printed material at room temperature. The reported method is a promising new strategy for advanced and easy manufacturing of complicated 3D structures.

I believe this work is timely and of great interest to the field of 3D printing and general polymer physics. The paper is well-written, and the scientific insights are clearly discussed. I recommend the publishing of this paper without delay. Here are a couple of minor comments and questions.

Response: We would like to express our gratitude to the reviewer for the positive comments on our work and for recommending its publication in Nature Communications. In the following section, we have provided a point-by-point response to the reviewer's comments.

Comments #1: The reported material showed outstanding stretchability before failure and good shape recovery performance. The reviewer is curious about the maximum elastic strain at different temperatures. Can the authors provide this information?

Response: We would like to thank the reviewer for this valuable comment. In response, we tested the maximum elastic strain of B1 and B2 at 80°C, as shown in the FigureR7. The remarkable 1500% elastic stretchability of B3 can be attributed to its abundant hydrogen bonding and loose crosslinking network in the organogel form. On the other hand, B1 and B2 have a higher crosslinking density, and the elevated temperature weakens the hydrogen bonding. However, they still exhibit relatively high elastic strain.

In response, we have added the following sentence in the Single vat g-DLP printing of materials with multi-property section:

“Both B1 and B2 can be stretched to more than 300% strain before failure at room temperature, and even higher elastic stretchability at elevated temperature (Supplementary Fig. 1).

Figure R7 (added as Supplementary Fig. 1). Strain-stress curves of B1 and B2 tested at 80 °C.

Comments #2: For the shape morphing demonstrations enabled by the mismatch strain, do the shapes change their configurations with time, i.e., due to stress relaxation of the viscous material? How long can the shapes be kept?

Response: We thank the reviewer for bringing up this valuable comment. For the typical hinge structure H13 we demonstrated in this work exhibited a good shape fixity under room temperature. The samples presented in the manuscript remained unchanged for several months without significant shape alterations.

In response, we have added the following sentence in the Shape-morphing with cold-draw

programming hinge section:

“Those shape-morphed structures can maintain their shape for months under room temperature without significant alterations.”

Comments #3: Typo in Figure 2c: Fixicty.

Response: We would like to express our appreciation to the reviewer for catching this typo. We have corrected the figure accordingly and conducted a thorough check of the manuscript to ensure the overall writing quality.

Comments #4: In Figure 2D, why is the nominal strain at the compressive portion of the hinge zero? It should be negative strain. The authors should double check or clarify. Do the FEA simulations in F and G share the same strain color bar as D?

Response: We thank the reviewer for pointing this out. The nominal strain in the longitudinal direction (stretching direction) should indeed be negative. We carefully checked the results and found that we mislabeled the legend of color map: the color in Figure 2D represents the maximum principal component of the nominal strain, instead of the “nominal strain” (without indicating component). Figure R8 shows the spatial distributions of the maximum principal strain (Figure R8A, same as that used in Figure 2D) and the longitudinal strain (Figure R8B). Figure R8B indeed shows the negative longitudinal strain in the compressive portion of the hinge. We would like to mention that the longitudinal strain is not directly available in Abaqus (FEA software we used), and to visualize it, here we built a cylindrical coordinate system “CSYS-1” whose “Axis 2” is along the angular direction, and the strain “NE22” can approximate the longitudinal strain for the hinge module. Due to the fact that such “NE22” may not exactly equal the longitudinal strain, we used maximum principal strain in our manuscript, which visualizes the longitudinal strain of the tensile portion, such as fibers.

Figure R8. Strain distributions of a representative deformed hinge that is shown in Figure 2D of the manuscript. Different components of the nominal strain are shown in (A-B). (A) The maximum principal strain, which is the same as that used in Figure 2D. (B) The longitudinal strain (in stretching direction). To visualize this strain component, we built a cylindrical coordinate system “CSYS-1” whose “Axis 2” is along the angular direction, and the strain “NE22” represents approximately the longitudinal strain for the hinge module. The results show negative longitudinal strain in the compressive portion of the deformed hinge.

We have corrected the legend of the color map and added more descriptions in the figure caption to emphasize that the maximum principal strain is used.

For Figure 2F and 2G, the color shows the von-Mises stress and is thus not based on the same color bar. To keep the color map consistent, we have replaced simulation snapshots with those sharing the same color bar (i.e., maximum principal nominal strain from 0 to 1.5) as Figure 2D.

REVIEWERS' COMMENTS

Reviewer #1 (Remarks to the Author):

The authors have addressed most of my comments, However, before publishing, a few issues still need clarification.

In response to comment #2, the authors have provided an explanation of the shape memory mechanism, which is commendable. However, I believe they should elaborate further. Specifically, the authors should mention the glass transition temperature, as this parameter plays a crucial role in fixing the temporary shape due to its higher value than room temperature. Additionally, the authors should emphasize how cross-linking contributes to the shape return capability by preventing permanent shape changes caused by sliding between polymer chains.

Regarding comment #3, the authors mention in their response that "The advanced feature of g-DLP is its ability to integrate different properties into a monolithic structure, greatly simplifying the fabrication of complex structures like the hinge structure presented in our work, where stiff fibers are embedded in a rubbery matrix." I believe this advantage should be prominently highlighted in the manuscript. In my opinion, cold programmability is primarily derived from the polymer's structural design. Even without 3D printing, simple UV light crosslinking with varying specific intensities can provide cold-programmable capabilities in the polymer samples.

For comment #7, the additional experiment is convincing. It would be better if the authors can add one corresponding printing/fabrication video in the process.

Reviewer #2 (Remarks to the Author):

My comments have been addressed adequately.

Reviewer #3 (Remarks to the Author):

The authors have addressed my comments and I recommend this paper for publishing in Nature Communications.

Responses to Reviewers

Dear reviewers,

Thank you for those insightful feedback and comments to help improve the quality of this manuscript.

Reviewer #1

The authors have addressed most of my comments, However, before publishing, a few issues still need clarification.

In response to comment #2, the authors have provided an explanation of the shape memory mechanism, which is commendable. However, I believe they should elaborate further. Specifically, the authors should mention the glass transition temperature, as this parameter plays a crucial role in fixing the temporary shape due to its higher value than room temperature. Additionally, the authors should emphasize how cross-linking contributes to the shape return capability by preventing permanent shape changes caused by sliding between polymer chains.

Response: We thank the reviewer for the comment. In response, we added following sentence in the corresponding results section:

“Both the chemical crosslinking by AUD and the hydrogen bonding generate crosslinking sites within the thermoset network to stabilize the polyacrylate chain segments and dictate the permanent shape of the network, contributing to such a high shape return capability.”

“As discussed earlier, the glassy SMPs B1 and B2 both possess a relatively high T_g above room temperature, which is critical to facilitate the fixing of the cold-programmed temporary shape.”

Regarding comment #3, the authors mention in their response that "The advanced feature of g-DLP is its ability to integrate different properties into a monolithic structure, greatly simplifying the fabrication of complex structures like the hinge structure presented in our work, where stiff fibers are embedded in a rubbery matrix." I believe this advantage should

be prominently highlighted in the manuscript. In my opinion, cold programmability is primarily derived from the polymer's structural design. Even without 3D printing, simple UV light crosslinking with varying specific intensities can provide cold-programmable capabilities in the polymer samples.

Response: We thank the reviewer for the suggestion. In response, we highlighted this in the discussion section.

“The advanced feature of g-DLP is its ability to integrate different properties into a monolithic structure, greatly simplifying the fabrication of complex structures like the hinges, where stiff fibers are embedded in a rubbery matrix. The g-DLP platform leverages the combination of cold programming and temperature-dependent shape-memory effects to create complex shape-morphing structures.”

For comment #7, the additional experiment is convincing. It would be better if the authors can add one corresponding printing/fabrication video in the process.

Response: We thank the reviewer for the suggestions. In response, we have added supplementary movie 8 to display the printing process of the hinge structures.

Reviewer #2

My comments have been addressed adequately.

Reviewer #3

The authors have addressed my comments and I recommend this paper for publishing in Nature Communications.